# Long-term effects of neonicotinoid insecticides on ants

Daniel Schläppi [1✉], Nina Kettler[1], Lars Straub[1,2], Gaétan Glauser[3] & Peter Neumann [1,2]

The widespread prophylactic usage of neonicotinoid insecticides has a clear impact on non-target organisms. However, the possible effects of long-term exposure on soil-dwelling organisms are still poorly understood especially for social insects with long-living queens. Here, we show that effects of chronic exposure to the neonicotinoid thiamethoxam on black garden ant colonies, *Lasius niger*, become visible before the second overwintering. Queens and workers differed in the residue-ratio of thiamethoxam to its metabolite clothianidin, suggesting that queens may have a superior detoxification system. Even though thiamethoxam did not affect queen mortality, neonicotinoid-exposed colonies showed a reduced number of workers and larvae indicating a trade-off between detoxification and fertility. Since colony size is a key for fitness, our data suggest long-term impacts of neonicotinoids on these organisms. This should be accounted for in future environmental and ecological risk assessments of neonicotinoid applications to prevent irreparable damages to ecosystems.

---

[1] Institute of Bee Health, Vetsuisse Faculty, University of Bern, Bern, Switzerland. [2] Agroscope, Swiss Bee Research Centre, Bern, Switzerland. [3] Neuchâtel Platform of Analytical Chemistry, University of Neuchâtel, Neuchâtel, Switzerland. ✉email: d.schlaeppi@mail.ch

The global decline of the entomofauna is alarming[1–3] and could have severe consequences for the functioning of natural ecosystems and food security[4,5]. While we heavily rely on agro-chemicals for food production, these products have also been identified as a driver of insect declines impairing ecosystem services[6–8]. Broad-spectrum insecticides like neonicotinoids are globally used to control pest species, but due to their non-specific mode of action, they also inevitably harm non-target organisms[9]. When applied to seeds, only a small fraction of the water-soluble neonicotinoid active ingredients are taken up by plants, whilst up to 95% remain in the environment and contaminate soils and water[10]. Widespread use of neonicotinoids led to ubiquitous contaminations of soils, even in areas that are supposedly neonicotinoid free[11].

Although environmental concentrations are often low, they may still induce sublethal effects on organisms. Most studies on sublethal effects have focused on pollinators due to their economic importance[12,13]. Impact on foraging behaviour, learning, orientation, memory abilities, immune functions, fertility, colony growth and reproductive output has been shown[10,14,15]. At a colony level, neonicotinoids contribute to the weakening of bees in complex manners[16,17]. However, the focus on pollination may only provide a partial picture. Indeed, pollination constitutes only a minor fraction of the value of ecosystem services[18]. Soils are essential sources for a wide diversity of ecosystem services[19], and soil invertebrates may be the best possible indicators of soil quality and thus the provided services[5]. Services such as decomposition, nutrient cycling and bioturbation provided by soil organisms are essential for sustainable agricultural production and are threatened by systemic insecticides[7].

Although data are still scarce, it is not surprising that adverse effects on non-target soil-dwelling invertebrates have been detected at individual and community level[20]. Subtle time-lag and time-cumulative effects are likely to occur due to persistent low concentrations of neonicotinoid insecticides in soils, yet detecting effects remains challenging[21]. While time cumulative toxicity has been reported for several species[22], including ants (*Linepithema humile*)[23], there are only very few long-term studies. For honey bees, long-term impacts of sublethal neonicotinoid exposure on colony performance and queen fate have been shown[17,24,25]. Such long-term consequences of chronic insecticide exposure are especially likely for long-lived organisms. Regarding long lifespans, ant queens are outstanding[26], especially black garden ant queens (*Lasius niger*) that can live to almost 30 years[27]. In addition to the exposure routes known for honey bees, namely direct exposure during application and indirect via food[28], ants can be chronically exposed to neonicotinoid residues in the soil. Like most ant species, *L. niger* can be regarded as sedentary because colonies are usually not mobile[29], potentially leading to exposure over decades. While short-term exposure of sublethal concentrations has already been shown to reduce reproductive success of solitary bees[30] and sperm traits of honey bees[14], very little is known about the uptake and detoxification of neonicotinoid residues over long time periods. In light of previous data on bees, trade-off scenarios between detoxification and reproduction appear likely. Even though there is evidence for lethal effects of neonicotinoids on ants at high concentrations[31], so far few studies have investigated sublethal and none long-term effects from chronic exposure at the colony level[32–34]. Furthermore, ants are not considered as representatives for soil-dwelling organisms and thus they are not covered in current risk assessments schemes by the EFSA or the OECD[35,36]. Also, the risk assessments for bees do not include the risk of exposure via neonicotinoid residues for soil nesting

hymenoterans[37,38]. There are requirements for testing parasitic wasps or other soil-dwelling non-target arthropods, but these organisms differ drastically in life-history traits and current tests only cover acute or short-term toxicity, but not long-term exposure. This is of great concern because ants play multiple crucial roles for ecosystem functioning[39]. As ecosystem engineers, they modify soil aggregation, structure and texture to establish optimal conditions for the colony[40]. This leads to increased porosity, improves aeration and hydraulic properties and thus helps to reduce erosion and runoff[41]. The high abundance of ants, import of food and the accumulation of faeces leads to enhanced nutrient cycling and accumulation of organic matter with higher concentrations of phosphorus, nitrogen and potassium[42–46]. Further, they contribute to essential ecosystem services such as pollination and natural pest control[47,48].

Here, we exposed gynes of *L. niger* captured after their nuptial flight to field-realistic sublethal levels of thiamethoxam, a common applied agricultural neonicotinoid insecticide. Colonies were monitored from foundation until the second overwintering to detect possible long-term effects on colony development (64 weeks). Further, we tested for the uptake and detoxification of neonicotinoids in different castes of eusocial insects by analysing the levels of thiamethoxam and its toxic metabolite clothianidin in queens and workers[49]. Both thiamethoxam and clothianidin are chlorothiazolylmethyl insecticides acting agonistically on insect nicotinic acetylcholine receptors[50]. However, thiamethoxam may be considered as a pro-insecticide having binding affinities up to 10,000-fold less potent than the other neonicotinoids, including clothianidin[51].

In this study, we show that chronic exposure to sublethal doses of thiamethoxam results in smaller colonies with fewer workers and larvae. Residue analysis of thiamethoxam and clothianidin further suggests that queens could have superior detoxification, with a possible trade-off scenario between reproduction and detoxification. Due to the importance of colony size, these results indicate that neonicotinoids might impact colony fitness and pose a threat to ecosystem functioning.

## Results

**Caste survival and body mass.** Until the end of the experiment (week 64), overall queen mortality was 20% and it was not significantly different amongst the three treatments (colonies chronically exposed to thiamethoxam: controls = 0 µg/L, low = 4.5 µg/L, high = 30 µg/L; each $N = 10$; log-rank test: $\chi^2_{(2)} = 0$, $p = 0.99$). On the fourth day after translocation to the nesting tube, one queen died (control) and another one died during the 3rd week (high treatment). A third queen (high treatment), which oviposited, but never produced viable offspring, died in week 50. All other queens ($N = 27$) were able to raise workers irrespective of treatment (chi-square test: $X^2_{(2)} = 2.22$, $p = 0.3292$), although three more queens died (week 42, control; week 47 and 59, both low treatments). Worker mortality appeared to be very low and was not traceable because the few dead bodies were deposed on waste piles[52]. Further, there were no significant differences in the duration of any developmental stages between the three treatments (Table 1).

At the end of the experiment, queen body mass did not differ between the three treatments (control: $33.11 \pm 5.98$ mg; low: $33.11 \pm 4.82$ mg; high: $35.14 \pm 4.24$ mg; each $N = 8$; ANOVA: $F_{2,21} = 0.426$, $p = 0.66$; Fig. 1a), but there were significant differences in the average worker body mass (ANOVA: $F_{2,21} = 7.97$, $p = 0.003$; Fig. 1b). The controls ($20.84 \pm 1.54$) were significantly heavier than the workers from high treatments ($17.46 \pm 1.85$ mg; Tukey post hoc test (tpht): $p = 0.002$,). Workers

**Table 1 Duration of developmental stages.**

| Thiamethoxam [µg/L] | Duration [days] ($Q_1$–$Q_3$) | | | | |
|---|---|---|---|---|---|
| | Preoviposition | Egg stage | Larval stage | Pupal stage | Ontogenesis |
| 0 | 1 [1;2] | 18 [18;19] | 8 [8;9] | 25 [24;26] | 52 [51;52] |
| 4.5 | 1 [1;1] | 17.5 [17;19] | 7.5 [7;9] | 24 [24;24.75] | 50 [48;52] |
| 30 | 1 [1;1] | 17.5 [17;18.25] | 8 [7.75;9] | 26 [24.5;27] | 51.5 [50.25;52] |
| Treatment comparison | Kruskal–Wallis test, $\chi^2_{(2)} = 2.602$, $p = 0.272$ | Kruskal–Wallis test, $\chi^2_{(2)} = 1.01$, $p = 0.603$ | Kruskal–Wallis test, $\chi^2_{(2)} = 0.27$, $p = 0.872$ | ANOVA, $F_{(2,24)} = 1.66$, $p = 0.603$ | Kruskal–Wallis test, $\chi^2_{(2)} = 1.35$, $p = 0.51$ |

Effect of thiamethoxam on the duration of each developmental stage in days shown as medians [1st 3rd quartiles]. Further, the test statistics to compare the three treatments (control = 0 µg/L, N = 9; low = 4.5 µg/L, N = 10; high = 30 µg/L, N = 8) are reported, revealing no statistically significant difference between the treatment groups.

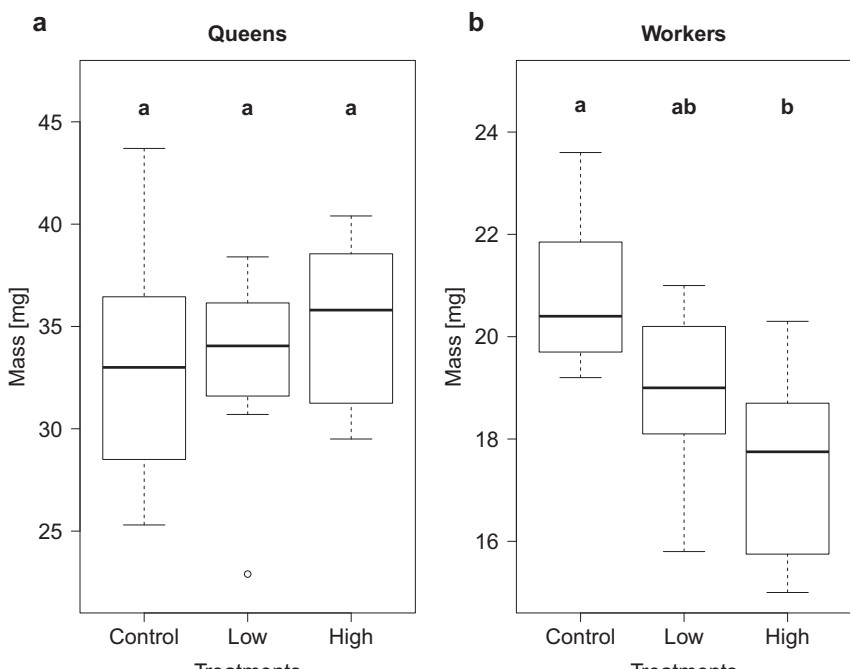

**Fig. 1 Thiamethoxam and body mass.** Mass (mg) of *Lasius niger* queens (**a**) and 20 pooled workers (**b**) at the end of the experiment (week 64) for the three thiamethoxam treatments (control = 0 µg/L, low = 4.5 µg/L, high = 30 µg/L; each N = 8). Boxplots are shown with the inter-quartile-ranges (box), medians (black line in box) and outliers (dots). Bold letters (a,b) indicate significant differences ($p < 0.05$) between treatments (Tukey post hoc test).

from low treatments (18.93 ± 1.69 mg) did not differ significantly from controls (tpht: $p = 0.08$) or high treatments (tpht: $p = 0.22$).

**Colony size**. Before the first overwintering (week 13; controls and high N = 9, low N = 10), there were no significant differences in the number of workers between the three treatments (ANOVA: $F_{2,25} = 0.78$, $p = 0.47$; Table 2, Fig. 2a). Further, there were no significant differences regarding the number of eggs, larvae or pupae (Table 2). However, before the second overwintering (week 64; each N = 8), there was a significant difference in the number of workers (ANOVA: $F_{2,21} = 16.03$, $p < 0.001$; Fig. 2b) and larvae (Kruskal–Wallis test: $\chi^2_{(2)} = 10.07$, $p < 0.01$). The controls had significantly more workers compared to both insecticide treatments (tpht: both $p$'s < 0.01), but there was no significant difference between the low and high treatment (tpht: $p = 0.33$). Further, the controls had more larvae compared to the high treatments (Dunn's test: $p = 0.026$), but there was no significant difference between the low treatment and the other two groups (Dunn's tests: control—$p = 0.06$; high—$p = 0.39$). Regarding the number of pupae and eggs at the end of the experiment, there

were no significant differences (Kruskal–Wallis tests: $\chi^2_{(2)} = 1.89$, $p = 0.38$; $\chi^2_{(2)} = 5.48$, $p = 0.06$).

**Neonicotinoid residues**. UHPLC-MS/MS analyses of pooled samples confirmed the absence (controls) as well as the presence and persistence of thiamethoxam in the treatment solutions of the low and high treatments (week 35: low = 6.05 µg/L, high = 31.77 µg/L; week 64: low = 5.02 µg/L, high = 19.5 µg/L). All controls were negative for both thiamethoxam and clothianidin (N = 8 queens, N = 8 workers). The concentration of thiamethoxam residues per gram dry mass in ants was significantly affected by the factor treatment (each N = 8; $\chi^2_{(1)} = 16.48$, $p < 0.001$), while neither the factor caste nor the interaction term (treatment:caste) showed significant effects ($\chi^2_{(1)} = 1.3$, $p = 0.25$; $\chi^2_{(1)} = 0.01$, $p = 0.92$; Fig. 3a). The concentration of clothianidin was significantly affected by treatment, caste and their interaction (caste—$\chi^2_{(1)} = 25.91$, $p < 0.001$; treatment—$\chi^2_{(1)} = 13.6$, $p < 0.001$; interaction—$\chi^2_{(1)} = 4.51$, $p = 0.03$; Fig. 3b). The clothianidin residue concentration was higher for samples exposed to higher concentrations, queens had lower concentrations compared to

**Table 2 Colony sizes.**

| Stage | Week | Thiamethoxam [µg/L] | | | Treatment comparison |
|---|---|---|---|---|---|
| | | 0 | 4.5 | 30 | |
| Workers | 13 | 18 [12;18] | 16.5 [13;19] | 16 [11;17] | ANOVA, $F_{2,25} = 0.78$, $p = 0.47$ |
| | 64 | 290.5 [230.75;306.75] | 188 [175;198.75] | 151.5 [137;184.75] | ANOVA, $F_{2,21} = 16.03$, $p < 0.0001$ |
| Eggs | 13 | 0 [0;0] | 0 [0;0] | 0 [0;2] | KW, $\chi^2_{(2)} = 4.01$, $p = 0.13$ |
| | 64 | 10 [5.25;18.75] | 3 [0;8.5] | 0 [0;3] | KW, $\chi^2_{(2)} = 5.48$, $p = 0.06$ |
| Larvae | 13 | 40 [36;44] | 35.5 [32;42] | 36 [34;39] | KW, $\chi^2_{(2)} = 2.51$, $p = 0.28$ |
| | 64 | 105.5 [63.75;150] | 49 [31.75;62] | 35.5 [23.75;47] | KW, $\chi^2_{(2)} = 10.07$, $p < 0.01$ |
| Pupae | 13 | 0 [0;0] | 0 [0;0] | 0 [0;0.75] | KW, $\chi^2_{(2)} = 4$, $p = 0.14$ |
| | 64 | 3 [0.75;7.25] | 2 [0.75;3.25] | 0.5 [0;2] | KW, $\chi^2_{(2)} = 1.89$, $p = 0.38$ |

Numbers of workers, eggs, larvae and pupae are shown as medians [1st 3rd quartiles]) before the first (week 13, controls and high $N = 9$, low $N = 10$) and second overwintering (week 64, each $N = 8$) for the three thiamethoxam treatments (control = 0 µg/L, low = 4.5 µg/L, high = 30 µg/L).

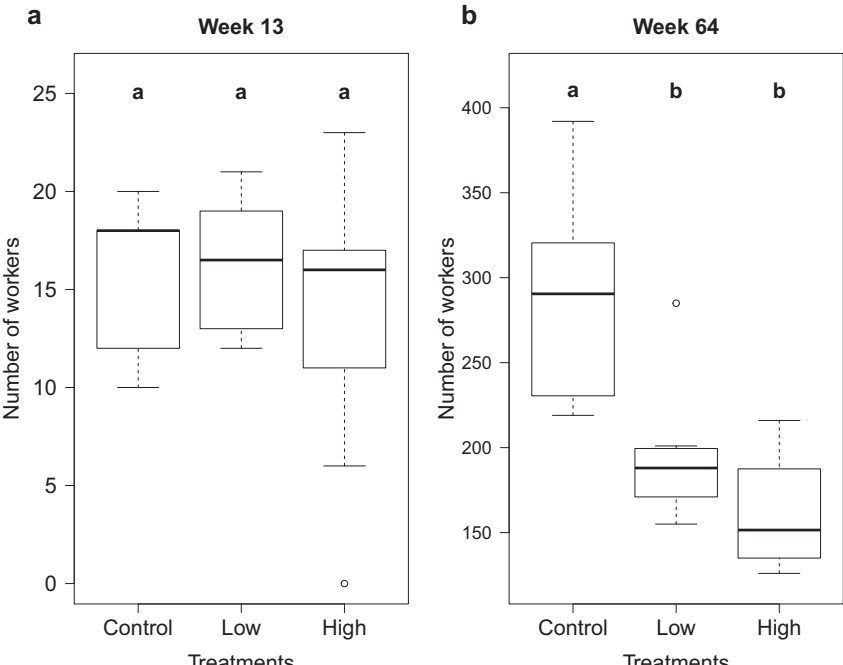

**Fig. 2 Thiamethoxam and colony size.** The number of workers in colonies for the three thiamethoxam treatments (control = 0 µg/L, low = 4.5 µg/L, high = 30 µg/L) before the first (**a**) and second (**b**) overwintering (week 13, controls & high $N = 9$, low $N = 10$; week 64, each $N = 8$). Boxplots are shown with the inter-quartile-ranges (box), medians (black line in box) and outliers (dots). Bold letters (a, b) indicate significant differences ($p < 0.05$) between treatments (Tukey post hoc test).

workers and the difference between castes increased with exposure concentration. Further, the clothianidin to thiamethoxam ratio was significantly higher in workers (median: 1.93, 1st–3rd quartile: 1.61–2.95) compared to queens (median: 0.55, 1st–3rd quartile: 0.42–0.77; $\chi^2_{(1)} = 9.1$, $p = 0.021$; Fig. 3c).

## Discussion

Our data clearly show long-term adverse effects of sublethal neonicotinoid exposure on ant colony sizes. While no effects were seen before the first overwintering, thiamethoxam-exposed colonies had fewer workers and larvae before the second winter. Further, queens had less of the toxic thiamethoxam metabolite clothianidin compared to workers suggesting superior detoxification, possibly contributing to low queen mortality, but resulting in a trade-off between detoxification and reproduction. Since social insects are essential for

terrestrial ecosystems and colony size is key for ant colony fitness, our results highlight the urgent need to improve the deployment and management of chemical pest control for more sustainable agriculture.

Queen mortality and colony failure were low for all treatments over the entire experiment, matching earlier findings with no effects of imidacloprid exposure on queen survival[34]. Likewise, the experimental duration did not exceed the natural lifespan of *L. niger* workers of up to 2 years and low worker mortality was found[53]. This implies that the field-realistic concentrations of thiamethoxam only have sublethal effects on ants. Nevertheless, small doses of thiamethoxam, which is metabolised to clothianidin[49], appear to have subtle impacts accumulating over time[54].

Only before the second overwintering, the smaller sizes of neonicotinoid-exposed colonies became apparent. These differences in the workforce are likely to become more pronounced over time, as a larger worker population is more efficient at

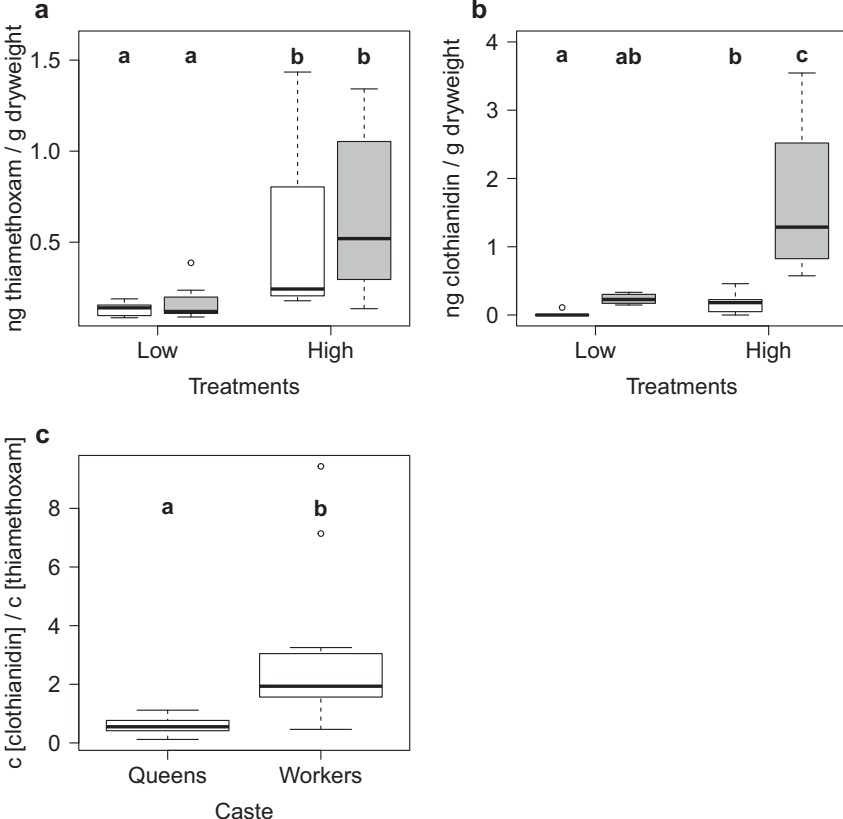

**Fig. 3 Neonicotinoid residues in ants.** Comparison of the residue concentration per gram dry mass of thiamethoxam (**a**) and clothianidin (**b**) found in ants (white boxplots = workers, grey boxplots = queens) for the two thiamethoxam treatments (low = 4.5 μg/L, high = 30 μg/L; each $N = 8$). **c** The concentration ratio of clothianidin/thiamethoxam for all samples positive for both clothianidin and thiamethoxam (queens $N = 7$, workers $N = 16$). Boxplots are shown with the inter-quartile-ranges (box), medians (black line in box) and outliers (dots). Bold letters (a, b) indicate significant differences ($p < 0.05$, linear mixed effect modelling with post hoc Tukey correction).

exploiting and dominating resources[55]. Before the second over-wintering, the control colonies were on average 38% and 56% larger than those in the low and high treatments. Further, colonies from the high treatments were smaller than the ones from the low treatment, but the difference was not statistically significant, indicating that an activated detoxification of queens compromises fertility regardless of dosage as long as lethal levels are not reached. Many factors of colony organisation and productivity are dependent on the number of workers[56,57]. An increased workforce is a competitive advantage, as inter-colony conflicts are usually won by the colony with a higher number of workers[58], especially during colony founding[59]. Most importantly, larger colonies have an earlier onset of the reproductive stage and produce more sexuals[60,61]. Considering the importance of colony size in the competitive context of ant communities, the observed effects are very likely of high relevance for colony performance and fitness in the field. Therefore, neonicotinoids affecting colony strength might also result in population declines even though the colony is the unit of reproduction. Also, other stress factors known to act together with insecticide exposure from other systems (e.g. pathogens[62], nutrition[63]) are absent from a laboratory setting, thereby constituting a conservative best-case scenario.

As expected, the neonicotinoid residue analyses showed that higher exposure concentrations resulted in higher loads of both thiamethoxam and its metabolite clothianidin. While the levels of thiamethoxam were not different between the castes, clothianidin and the ratio of clothianidin to thiamethoxam were both lower in queens. These differences between workers and queens can be either explained by different routes and amounts of exposure or differences in their abilities to detoxify the toxicants. The uptake of neonicotinoids by queens might be either direct via topical exposure in the nesting tubes or indirect via trophallactic fluid from her workforce. Trophallactic fluids are used for sharing food and play a role in communication[64]. In contrast to nursing and royal jellies produced by bees, which are virtually free from toxics, such as alkaloids protecting bee larvae[65], trophallactic fluids come from the crop and are only partly pre-processed and predigested[53]. Because the neonicotinoid breakdown in the workers appears not to be sufficient to reduce the levels of clothianidin entirely, we can assume that higher levels of clothianidin should reach the queens if thiamethoxam were preprocessed. However, the residue analyses show that clothianidin levels in queens are almost neglectable. Furthermore, the residue levels of thiamethoxam indicate equal exposure to the castes. Therefore, the low concentrations of clothianidin found in queens indicate superior detoxification. Indeed, it has been shown that queens can be more tolerant of toxins compared to workers[66]. However, this conclusion is only valid if clothianidin is the only toxic metabolite of thiamethoxam in ants. According to metabolism studies based on total radioactive residue values, clothianidin is the only relevant metabolite for bees when applied as a seed treatment[67]. N-desmethylthiamethoxam, another metabolite of thiamethoxam, which has also been reported as a nicotinic agonist, seems to be formed only in very limited amounts from thiamethoxam and

the observed low concentrations are not likely to contribute to insecticidal potency[49]. However, in ants, the metabolic fate of thiamethoxam has not yet been explored and we therefore cannot exclude potential toxic effects of other intermediates.

Being able to better cope with thiamethoxam could also explain that no differences regarding body mass were detected in queens, whereas workers exposed to thiamethoxam were on average lighter compared to the controls. That larval exposure to neonicotinoids can affect adult body size has been shown in butterflies[68]. A reduced worker body mass could be a disadvantage for colonies, as it can be important for several fitness-relevant aspects[69]. Since queens acquire food from workers, it might be expected that lower worker body mass also results in reduced queen body weight. However, our results suggest that worker body mass did not affect the queen body mass. Reproductive division of labour in social insects selects for differential resource-allocation[70], i.e. favouring protein provisioning of queens. This may well explain why they are well supplied even under worse conditions[71].

Despite the pronounced physiological differences between queens and workers, there is little known on the underlying mechanisms that could explain differences in detoxification abilities. Possible candidates are cytochrome P450 mono-oxygenases, esterases and glutathione transferases, three groups of detoxifying enzymes that have been shown to play a role in insecticide resistance[72,73]. Cytochrome P450 monooxygenases especially represent a key detoxification mechanism of neonicotinoids in insects[73,74]. Further, queens have a larger fat body, sometimes compared to the vertebrate liver, which plays an essential role in many vital functions, including detoxification[75], though this might not be of importance for highly polar neonicotinoids that are unlikely to accumulate in fats. In light of the reproductive division of labour as a cornerstone of the biology of eusocial insects[70] and superorganism resilience due to workers being replaceable units[76], it seems evident that any traits, allowing queens to cope better with stressors and thus to live longer will be favoured by natural selection. Even though queens of social insects have overcome the trade-off between longevity and somatic maintenance[77], the underlying mechanisms explaining their longevity are still poorly understood[78]. Our data suggest that one mechanism appears to be superior detoxification by queens. It appears as if queens are facing a typical trade-off scenario between activation of detoxification or immune pathways and reproduction[79,80].

Based on our results, the most likely mechanisms for the reduced colony size are either an impaired development of the brood, an altered behaviour of the workers or reduced queen fertility. Wang, et al.[34] showed that at high concentrations of imidacloprid (250 µg/L) the time to larval emergence was delayed and no pupae or adult workers were produced. However, no difference in the developmental time of the brood and no developmental abnormalities were observed at the used field-realistic concentrations of thiamethoxam. We cannot exclude that thiamethoxam might have affected the fertility of the queens in the second year, e.g. long-term exposure to neonicotinoids could lead to decreased sperm quality[14]. On the other hand, compromised foraging behaviour and nursing[33] could also be reasons for the slowed development of the colonies and reduced worker body mass in the treatments.

Accumulating subtle long-term impacts of neonicotinoids on ants are alarming. Our results indicate that neonicotinoids contribute to the weakening of ant colonies in the long run and thus can pose a threat to ecosystem functioning. This study is an example of an effect from an environmental contaminant with subtle sublethal effects on long-lived organisms, that become visible only after long-term monitoring, but with possibly far-reaching consequences.

To fully understand the threat of toxic substances in risk assessments, long-term studies covering full life-cycles are therefore required to determine the ecological risk, especially for long-lived organisms like social insect colonies. To prevent irreparable damages to functioning ecosystems, we therefore suggest to either fully incorporate long-term effects in risk assessment schemes, or to make a shift in plant protection strategies and apply the precautionary principle when making policy decisions.

## Methods

**Experimental setup**. Three treatments were established to test for the effects of chronic exposure of thiamethoxam. Two concentrations of thiamethoxam reflecting previously reported field-realistic levels (4.5 µg/L = treatment low; 30 µg/L = treatment high) were selected and distilled water for the controls[81,82]. Thiamethoxam (>99.9% purity, Sigma-Aldrich, St. Louis, Missouri, USA) was dissolved directly in distilled water. Solutions were freshly prepared at the beginning of the experiment and after overwintering. A sterile cotton wool ball separated nesting tubes (Supplementary Fig. 1; 155 mm length, 14 mm inner diameter) into two compartments. The rear chamber was filled with one of the three treatment solutions (10 ml). The first cell housed the queen and was closed by another cotton wool plug. Nesting tubes were wrapped in aluminium foil, maintained at RT (19–23 °C) and protected from sunlight.

Gynes of *Lasius niger* were collected in Bern, Switzerland (30.07.2016, week 0; Supplementary Fig. 2) after their nuptial flights and tested for fertility by keeping them individually in light-protected beakers for 12 days, with water provided ad libitum. Upon initiation of the experiment, a total of 30 individual laying ant queens were randomly chosen and allocated to one of the three treatment groups (each N = 10). The colonies were overwintered in a fridge at 6 °C from week 15 to 32 weeks with a 2-week acclimatisation period at 14 °C before and after overwintering. Week 34, the colonies were transferred into new nesting tubes with freshly prepared treatment solutions and an attached foraging arena (135 × 68 × 32 mm). After 64 weeks, the experiment was ended prior to the second overwintering. All colonies were frozen at −80 °C and the exact number of adults and brood was recorded per colony. Further, queens and 20 randomly pooled workers were weighted per colony, to see whether there might be an effect on body mass. To confirm the persistence of thiamethoxam over time, pooled samples of the neonicotinoid solutions from the nesting tubes were collected at the moment of nest translocation and the end of the experiment, stored at 4 °C and checked with ultra-high performance liquid chromatography-tandem mass spectrometry (UHPLC-MS/MS).

Until the first overwintering, colony development (number of eggs/larvae/pupae/adults) was recorded daily for 50 days and thereafter every second day and we derived life-stage durations for eggs, larvae and pupae as well as the pre-oviposition time[34]. Before overwintering the first time, once the first workers emerged, colonies were provided twice with 30 µl droplets of sugar-water (50% mass fraction of sugar). After overwintering, the ants were provided weekly with a sugar-water drenched cotton ball and drosophila flies (*Drosophila hidey*). In July 2016, we switched from *Drosophila* to honey bee (*Apis mellifera*) pupae as a protein source to satisfy the increased protein needs of the colony.

**Neonicotinoid analyses**. To confirm the uptake of neonicotinoids by the ants, we tested queens and workers from all colonies that lasted until the end of the experiment for thiamethoxam and its metabolite clothianidin using UHPLC-MS/MS analysis. Abdomens from queens (24.75 ± 4.52 mg fresh mass) and pooled samples with worker abdomens (25.02 ± 0.23 mg fresh mass) were prepared using an adapted QuEChERS protocol[11]. The samples were weighed in 2 ml tubes and freeze-dried overnight in a FreeZone lyophilizer (Labconco). The samples were ground for 6 min in a Retsch 400 MM tissue lyser at 27 Hz using five chrome metal beads (3 mm), quickly centrifuged, provided with 1.5 ml of acetonitrile and 8 µL of internal standard solution (IS, 125 ng/ml in MeOH, containing the two labelled neonicotinoids thiamethoxam and clothianidin) and shaken for 5 min using the Retsch 400 MM tissue lyser at 30 Hz. The resulting solution was centrifuged (14000 g, 3 min) and then as much supernatant as possible was transferred into a 15 ml Falcon tube containing 3.25 g of extraction salts (2 g $MgSO_4$, 0.5 g NaCl, 0.5 g sodium citrate dihydrate and 0.25 g sodium citrate sesquihydrate). The pellet was re-extracted with 1.5 ml acetonitrile and the supernatants combined. The further purification was performed according to Humann-Guilleminot, et al.[11] with the centrifugations at 3200 g for 3 min.

Neonicotinoid analyses were performed on an Acquity UPLC system (Waters, Milford, MA) coupled to a TQ-S triple quadruple (Waters)[83]. Quantification of thiamethoxam and clothianidin was performed by internal calibration using solutions in MeOH 25% at 0.005, 0.05, 0.5, 5 and 15 ng/ml, each containing internal standards (concentration of 5 ng/ml). Linear regressions weighted by 1/x were applied. Concentrations were normalised to the mass of the samples and expressed in ng/g dry mass to compare queens and workers. Detection limits of

quantification for thiamethoxam and clothianidin were 20 and 40 pg/g of tissues, respectively. Blank samples were included as negative controls.

**Statistics and reproducibility**. All statistical analyses were performed using R version 3.5.1[84]. Data or models were checked for normal distribution with the Shapiro-Wilk test and homogeneity of variances with the Levene's test and subsequent statistical tests chosen accordingly. Unless specified differently each colony was used as a replicate. The moment a queen died, the respective colony was removed from subsequent analyses, resulting in the sample sizes mentioned in the results section.

To test for differences in queen survival between the three groups, a log-rank test was performed using the *survdiff* function within the R package survival[85]. Further, the queen's ability to raise workers successfully depending on the treatment was assessed by a Pearson's Chi-squared test. To test for differences in body mass of queens and workers between the treatments, we used an ANOVA. In case of significance, Tukey HSD post hoc tests were applied to test for pairwise differences. The duration of life stages (eggs, larvae, pupae and complete ontogenesis) was tested with a Kruskal–Wallis Test or an ANOVA. To compare the number of workers between the three treatments we used linear models and visually inspected the residual plots for deviations from homoscedasticity and normality.

Additionally, we compared the amount of brood in the nests before overwintering and at the end of the experiment. We proceeded with all the brood analyses as we did when comparing the duration of life stages. In the case of a significant Kruskal–Wallis test, the pairwise comparison for was done with the Dunn's test for multiple comparisons with Bonferroni correction from the R package dunn.test[86].

We used the *lmer* function from the package lme4[87] to perform a linear mixed effect analysis of the relationship between ant caste and neonicotinoid residues (thiamethoxam or clothianidin). The controls were all negative, confirming that there was no detectable contamination prior to the experiment and thus they were neglected in the following models. For each substance, the concentration per gram dry mass entered the model as a response variable. As fixed effects, we used caste and treatment, including their interaction term. As a random effect, we added the colony identity to control for the fact that we tested both the queen and workers from each colony. To meet the model assumptions for clothianidin, a small positive constant (half of the lowest observed value) was added to values that were zero and then the response variable was transformed using a reciprocal transformation ($1/x$). Visual inspection of residual plots of the models did not reveal deviations from homoscedasticity or normality. *P*-values were obtained by comparing the model with the effect in question against the model without it, using an ANOVA as a likelihood ratio test. Post hoc pairwise comparison was performed with *p*-values adjusted according to Tukey to address the problem of multiple comparisons. For all samples which were positive for both clothianidin and thiamethoxam (queens $N = 7$, workers $N = 16$) we calculated the ratio of clothianidin to thiamethoxam and then compared compare it between the two castes with linear mixed-effect modelling using caste as fixed and colony identity as a random factor and proceeded like above.

**Reporting summary**. Further information on research design is available in the Nature Research Reporting Summary linked to this article.

## Data availability

Source data underlying plots shown in figures are available in Supplementary Data 1. All other data (if any) are available upon reasonable request.

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

## Acknowledgements

We wish to thank Kaspar Roth and Laurence Lachat for technical support and Laura Bosco for editing a previous version of the manuscript. Financial support was provided by the Béatrice Ederer-Weber Foundation to P.N., the Vinetum Foundation to P.N. and the Bundesamt für Umwelt to L.S. and P.N.

## Author contributions

D.S. and P.N. conceived and designed the study; D.S. and N.K. conducted the research and performed the experiments with inputs from L.S. and P.N.; P.N. provided laboratory space and materials; G.G. provided know-how, laboratory space and materials for the UHPLC-MS/MS neonicotinoid analysis; D.S. and N.K. analyzed the data; D.S. wrote the paper with contributions from all authors. All authors edited and approved the paper.

## Competing interests

The authors declare no competing interests.

## Additional information

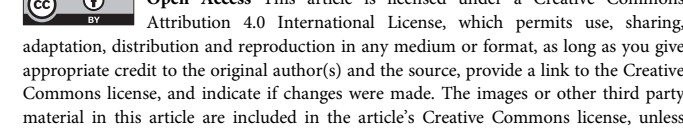

