## [Peer Review File · Communications Biology]

Reviewers' comments:

Reviewer #1 (Remarks to the Author):

Neonicotinoid insecticides have been reported for their potentially harmful effects on pollinating insects such as bees. Their chronic effects on other insects has also been reported, including ants. This research demonstrated a long-term (two seasons) effect of thiamethoxam on black garden ant colonies. It is alarming!!!. Since thiamethoxam has been used as an active ingredient in ant bait, this research also provides new perspectives on the toxicity of this neonicotinoid to ants. I don't have problems with experimental design and the data collection and analysis. However, I do have problem with one major conclusion. I believe that this research clearly shows the long-term adverse effects of sublethal thiamethoxam on ants; however, the conclusion on queen's superior detoxification is not very solid. This conclusion is based on that queens and workers differed in the residue-ratio of thiamethoxam to its metabolite clothianidin. The conclusion is valid if the clothianidin is the ONLY toxic thiamethoxam metabolite in ants. I do not know whether it is the case. Authors may need to at least discuss this in the manuscript.

In introduction, it will be nice if authors can extend a little bit more to comment on ants as ecosystem engineers. For example, certain ant species are also important pollinators. Authors may need to provide a little more information on clothianidin. How is the clothianidin related to thiamethoxam?, such as their chemical structures and toxicities to insects.

Ln 78: Colonies were monitored for two developmental seasons until the second overwintering to detect possible long-term effects on colony development. The word season is quite vague. Please define the developmental season.

Ln 95. At the end of the experiment, queen-body mass did not differ significantly between the three treatments but there were significant differences in the average worker body mass. It seems worker body mass did not affect the queen-body mass. Since queens acquire food from workers, it is expected that less workers may result in reduced queen body weight. Authors needed to discuss this in the Discussion Section.

Ln 262: "Once the first workers emerged in the first season, colonies were provided twice with 30 µl droplets of 1:1 sugar-water. In the second season, the ants were provided weekly with a 1:1 sugar-water drenched cotton ball and drosophila flies (*Drosophila* hidey). In July 2016, we switched from *Drosophila* to honeybee (*Apis mellifera*) pupae as a protein source to satisfy the increased protein needs of the colony". Why no protein was provided in the first season? Is it normal to an ant colony in the field?

Reviewer #3 (Remarks to the Author):

Excellent design and dataset. This work complements similar work done with honey bees colonies (Sandrock et al. 2014, PLoS One 9, e103592; Dively et al. 2015, PLoS One 10, e0118748) and communities of aquatic arthropods (Hayasaka et al. 2012, Ecotoxicol. Environ. Saf. 80, 355-362), all of which demonstrate the subtle impairments that neonicotinoid insecticides have on organisms and communities over time periods longer than a year.

The manuscript is well written and structured. However, the number of replicated colonies per treatment is not mentioned in the description of results, Tables and Figures, although it can be deduced from the experimental setup; I recommend to include it (N = 10). Only minor comments require attention, as indicated below.

Specific comments

Line 21: "...a superior detoxification system."

Line 24: "our data"

Line 42: "...sublethal effects on organisms."

Line 55: the adjective is 'chronic' not 'chronical'

Lines 56-8: agree that is not easy to detect those time-cumulative effects, but a recent paper by Sanchez-Bayo and Tennekes (2020, *Int. J. Environ. Res. Pub. Health* 17, 1629) shows that such cumulative effects have also been described for Argentine ants; it might be worth pointing this out here.

Line 72: it is true that no 'soil nesting hymenopterans' are required by the regulators worldwide; however, there is a requirement for testing parasitic wasps, and some of the non-target arthropods are soil organisms (i.e. the predatory mite *Hypoaspis aculeifer*, OECD test 226). However, those tests are only acute or short-term toxicity tests and do not cover the long-term exposure that the authors present in this manuscript.

Line 74: indeed, and a good example of their engineering benefits to soil ecosystems is that ants help infiltration and thus reduce erosion and runoff – see Cerdà and Jurgensen, 2008. *J. Appl. Entomol.* 132, 306-314.

Line 76: industrial? Isn't thiamethoxam a registered agricultural pesticide? Suggest replacing industrial with agricultural.

Line 139: "...size is key for ant..."; delete 'a'

Line 141: I don't follow your argument here. What is meant by 'sustainable plant protection' in this context? Less damaging insecticides? Better regulation of chemicals? Please clarify this sentence.

Line 152: it is hard to see these differences in Fig. 3D (by the way, this figure is not mentioned anywhere in the text). In terms of worker numbers the difference between control and low treatment colonies is more like 5%, and for the high treatments about 20%. In terms of larvae numbers (Fig. 3B), the difference is approx. 12% for the low treatment and 20% for the high treatment. Can you check this point?

Line 194: a more recent reference on this issue is the review by Matsuda et al. 2020, *Annu. Rev. Pharmacol. Toxicol.* 60, 241-255.

Lines 218-20: the authors have shown a 'weakening of ant colonies' and discussed the effects that a lower number of workers may have on ecosystem functioning. However, as eusocial insects, the survival of any ant species depends on the survival of the colony; it is the colony that matters, and as long as queens keep forming colonies, albeit of smaller worker size, ant species will not experience population declines, unlike many other insects. It may be worth to comment on this point.

Line 244: nowhere in the manuscript I could find the number of colonies used in this experiment. Here the authors mention '30 individual laying ant queens were randomly chosen and allocated to one of the three treatment groups', which works out as 10 queens per treatment, correct? The number of colonies (N) in the experiment should be included in Tables 1 and 2 and Figures 1 and 2.

Line 251: I assume the authors weighed the queens individually, so what kind of scale was used? It has to be very sensitive!

Line 261: how was this rough estimation done? Provide a simple description please.

Response to comments of the Editors

We ask you to address why you chose clothianidin as a major toxic thiamethoxam metabolite in ants.

Response: This is indeed an important aspect, which has also been pointed out by the first reviewer that needs to be addressed. Therefore we added a section to the introduction and the discussion (Lines 93ff & 200ff; see comments 2&4 of reviewer 1).

According to the European Food Safety Authority which performed metabolism studies based on total radioactive residue values, clothianidin is the only relevant metabolite of thiamethoxam for bees when applied as seed treatment [1]. N-desmethylated thiamethoxam, another metabolite of thiamethoxam, which was also reported as a nicotinic agonist, seems to be formed only in very limited amounts by insects and the observed low concentrations are not likely to contribute to insecticidal potency [2]. Therefore, on the basis on the existing literature, we decided to focus only on clothianidin. However, we now also discuss that in ants, the metabolic fate of thiamethoxam has not been explored and we cannot fully exclude certain potential toxic effects of other intermediates.

[1] European Food Safety Authority (EFSA). Peer review of the pesticide risk assessment for bees for the active substance thiamethoxam considering the uses as seed treatments and granules. *EFSA Journal*, 2018, 16. Jg., Nr. 2, S. e05179.

[2] Nauen, R. et al. Thiamethoxam is a neonicotinoid precursor converted to clothianidin in insects and plants. *Pesticide Biochemistry and Physiology*, 2003, 76. Jg., Nr. 2, S. 55-69.

Response to comments of Reviewer 1

1. Neonicotinoid insecticides have been reported for their potentially harmful effects on pollinating insects such as bees. Their chronic effects on other insects has also been reported, including ants. This research demonstrated a long-term (two seasons) effect of thiamethoxam on black garden ant colonies. It is alarming!!!. Since thiamethoxam has been used as an active ingredient in ant bait, this research also provides new perspectives on the toxicity of this neonicotinoid to ants. I don't have problems with experimental design and the data collection and analysis.

Response: We are delighted by this positive comment.

2. However, I do have problem with one major conclusion. I believe that this research clearly shows the long-term adverse effects of sublethal thiamethoxam on ants; however, the conclusion on queen's superior detoxification is not very solid. This conclusion is based on that queens and workers differed in the residue-ratio of thiamethoxam to its metabolite clothianidin. The conclusion is valid if the clothianidin is the ONLY toxic thiamethoxam metabolite in ants. I do not know whether it is the case. Authors may need to at least discuss this in the manuscript.

Response: The reviewer has a valid point. We appreciate this comment and now discuss the matter in the manuscript (lines 200ff). According to the European Food Safety Authority which performed metabolism studies based on total radioactive residue values, clothianidin is the only relevant metabolite of thiamethoxam for bees when applied as seed treatment [1]. N-desmethylated thiamethoxam, which was also reported as a nicotinic agonist, seems to be formed only in very limited amounts by insects and the observed low concentrations are not likely to contribute to insecticidal potency [2]. Therefore, on the basis on the existing literature, we decided to focus only on clothianidin. However, it should be noted that in ants, the metabolic fate of thiamethoxam has not been explored and we cannot fully exclude certain potential toxic effects of other intermediates.

[1] European Food Safety Authority (EFSA). Peer review of the pesticide risk assessment for bees for the active substance thiamethoxam considering the uses as seed treatments and granules. *EFSA Journal*, 2018, 16. Jg., Nr. 2, S. e05179.

[2] Nauen, R. et al. Thiamethoxam is a neonicotinoid precursor converted to clothianidin in insects and plants. *Pesticide Biochemistry and Physiology*, 2003, 76. Jg., Nr. 2, S. 55-69.

3. In introduction, it will be nice if authors can extend a little bit more to comment on ants as ecosystem engineers. For example, certain ant species are also important pollinators.

Response: Thank you for this constructive suggestion. We now dedicate a section (lines 78ff) to address the activity of ants as ecosystem engineers. Pollination was included as ecosystem service (lines 84-85), because it is not considered ecosystem engineering [1].

[1] Gutiérrez, JL. & Jones, CG (2008) Ecosystem engineers. In: *Encyclopedia of Life Sciences*, John Wiley & Sons Ltd, Chichester. <http://www.els.net> [doi: 10.1002/9780470015902.a0021226]

4. Authors may need to provide a little more information on clothianidin. How is the clothianidin related to thiamethoxam?, such as their chemical structures and toxicities to insects.

Response: This is a good suggestion. We added a section on clothianidin and thiamethoxam and how they are related (Lines 93ff).

5. Ln 78: Colonies were monitored for two developmental seasons until the second overwintering to detect possible long-term effects on colony development. The word season is quite vague. Please define the developmental season.

Response: We agree with the reviewer. The term developmental season was not adequately defined. We defined it as the period which covers two times the span between spring and late summer, during which the colonies produce brood and grow in terms of worker numbers even though we did not cover two full years. To improve legibility, we changed our wording throughout the manuscript and no longer use the term.

6. Ln 95. At the end of the experiment, queen-body mass did not differ significantly between the three treatments but there were significant differences in the average worker body mass. It seems worker body mass did not affect the queen-body mass. Since queens acquire food from workers, it is expected that less workers may result in reduced queen body weight. Authors needed to discuss this in the Discussion Section.

Response: We appreciate that the reviewer pointed this out and it is now discussed (Lines 212ff).

7. Ln 262: "Once the first workers emerged in the first season, colonies were provided twice with 30 μ l droplets of 1:1 sugar-water. In the second season, the ants were provided weekly with a 1:1 sugar-water drenched cotton ball and drosophila flies (*Drosophila* hidey). In July 2016, we switched from *Drosophila* to honeybee (*Apis mellifera*) pupae as a protein source to satisfy the increased protein needs of the colony". Why no protein was provided in the first season? Is it normal to an ant colony in the field?

Response: *Lasius niger* (and many other species) have claustral colony founding, meaning that after excavating their nest, new queens seal it up and raise their first brood by metabolizing nutritional reserves without going outside to forage [1]. Once, the first workers arrive they start to take over duties in the nest including foraging. Consequently, until workers emerge, our laboratory feeding approach reflects the natural development. We therefore chose to not provide protein food in the first season.

[1] Brown, M. J., & Bonhoeffer, S. (2003). On the evolution of claustral colony founding in ants. *Evolutionary Ecology Research*, 5(2), 305-313.

Response to comments of Reviewer 3

1. Excellent design and dataset. This work complements similar work done with honey bees colonies (Sandrock et al. 2014, PLoS One 9, e103592; Dively et al. 2015, PLoS One 10, e0118748) and communities of aquatic arthropods (Hayasaka et al. 2012, Ecotoxicol. Environ. Saf. 80, 355-362), all of which demonstrate the subtle impairments that neonicotinoid insecticides have on organisms and communities over time periods longer than a year. The manuscript is well written and structured.

Response: We are very glad about these positive comments. The Sandrock et al. 2014 paper was indeed mentioned in our manuscript. Further, we added an additional sentence (line 58) where we now specifically mention long-term effects in bees and cite both the Sandrock and the Dively paper.

2. However, the number of replicated colonies per treatment is not mentioned in the description of results, Tables and Figures, although it can be deduced from the experimental setup; I recommend to include it (N = 10).

Response: Thank you for pointing this out. The sample size is now mentioned in the results, tables and figures.

3. Only minor comments require attention, as indicated below.

Specific comments:

Line 21: "...a superior detoxification system."

Response: We agree with the reviewer and adjusted the sentence.

4. Line 24: "our data"

Response: We agree with the reviewer and adjusted the sentence.

5. Line 42: "...sublethal effects on organisms."

Response: We agree with the reviewer and adjusted the sentence accordingly.

6. Line 55: the adjective is 'chronic' not 'chronical'

Response: Thanks for pointing this out. The mistake has been corrected.

7. Lines 56-8: agree that is not easy to detect those time-cumulative effects, but a recent paper by Sanchez-Bayo and Tennekes (2020, Int. J. Environ. Res. Pub. Health 17, 1629) shows that such cumulative effects have also been described for Argentine ants; it might be worth pointing this out here.

Response: The suggested review and the paper on Argentine ants are indeed worth pointing out. Both references have been included in the introduction (Line 56ff).

8. Line 72: it is true that no 'soil nesting hymenopterans' are required by the regulators worldwide; however, there is a requirement for testing parasitic wasps, and some of the non-target arthropods are soil organisms (i.e. the predatory mite *Hypoaspis aculeifer*, OECD test 226). However, those tests are only acute or short-term toxicity tests and do not cover the long-term exposure that the authors present in this manuscript.

Response: Thanks for pointing this out. We added one additional sentence to the introduction to mention this point (lines 75ff).

9. Line 78: indeed, and a good example of their engineering benefits to soil ecosystems is that ants help infiltration and thus reduce erosion and runoff – see Cerdà and Jurgensen, 2008. J. Appl. Entomol. 132, 306-314.

Response: Reviewer 1 pointed out as well that we should describe ecosystem engineering by ants in a bit more detail. Thus, we dedicated several lines to this (Lines 79ff) and also included the suggested study by Cerda and Jurgensen (2008).

10. Line 76: industrial? Isn't thiamethoxam a registered agricultural pesticide? Suggest replacing industrial with agricultural.

Response: We agree with the reviewer's view and have changed the word accordingly.

11. Line 139: "...size is key for ant..."; delete 'a'

Response: Thank you for pointing this out. The phrase was corrected.

12. Line 141: I don't follow your argument here. What is meant by 'sustainable plant protection' in this context? Less damaging insecticides? Better regulation of chemicals? Please clarify this sentence.

Response: We have reworded the sentence and hope that it clearer now (lines 154ff)

13. Line 152: it is hard to see these differences In Fig. 3D (by the way, this figure is not mentioned anywhere in the text). In terms of worker numbers the difference between control and low treatment colonies is more like 5%, and for the high treatments about 20%. In terms of larvae numbers (Fig. 3B), the difference is approx. 12% for the low treatment and 20% for the high treatment. Can you check this point?

Response: In line 152 (now 168) we refer to the colony size (number of workers), at the end of the experiment before the second overwintering. These differences are visualized in Figure 2B. The figures 3A-D only show the colony development during the first season. We hope that clarifies the issue.

Thanks for pointing out that the figure was not mentioned in the text. The figure showed the colony development from foundation until the first overwintering. However, this information is not crucial for the message of this paper and also redundant in light of figure 2A. Consequently, we see no gain or even a distraction by keeping this figure and we have decided to delete it from the manuscript. Figure numbers were adjusted accordingly.

14. Line 194: a more recent reference on this issue is the review by Matsuda et al. 2020, Annu. Rev. Pharmacol. Toxicol. 60, 241-255.

Response: This is a very good suggestion. The review is now cited (line 222).

15. Lines 218-20: the authors have shown a 'weakening of ant colonies' and discussed the effects that a lower number of workers may have on ecosystem functioning. However, as eusocial insects, the survival of any ant species depends on the survival of the colony; it is the colony that matters, and as long as queens keep forming colonies, albeit of smaller worker size, ant species will not experience population declines, unlike many other insects. It may be worth to comment on this point.

Response: We totally agree with the reviewer that the colony is the unit of reproduction in social insects. Because of this, we discuss the impact of colony size on colony fitness (lines 172ff). However, most importantly smaller colonies are less competitive and hence likely to produce less sexuals. Therefore, neonicotinoids affecting the colony strength might nevertheless result in population declines. To make this clearer we added one more sentence to the discussion (line 179).

16. Line 244: nowhere in the manuscript I could find the number of colonies used in this experiment. Here the authors mention '30 individual laying ant queens were randomly chosen and allocated to

one of the three treatment groups', which works out as 10 queens per treatment, correct? The number of colonies (N) in the experiment should be included in Tables 1 and 2 and Figures 1 and 2.

Response: The sample size / number of colonies have now been added to the results, tables and figures. Indeed, we start with a sample size of N=10 per treatment. However, the sample size decreases over time because once the queen of a colony died, the respective colony was removed from further analyses (mentioned in the material and methods section, line 323). The sample size at the end of the experiment is N=8 per treatment.

17. Line 251: I assume the authors weighed the queens individually, so what kind of scale was used? It has to be very sensitive!

Response: Yes, the queens were weighed individually. For this purpose, we used an analytical balance of Mettler Toledo: Mettler Toledo AT400 with a precision of 0.1 mg.

18. Line 261: how was this rough estimation done? Provide a simple description please.

Response: The estimation was done as follows: First, the aluminium foil was removed. Then workers in the arena followed by workers in the nesting tube were counted with a tally pocket counter (always by the same person). Because workers tend to clump together and pile up with increasing colony size, the estimate is more inaccurate the bigger colonies get (colony size tended to be underestimated). However, for the analyses only the exact final counts before overwintering were used.

Because we only use the exact counts before overwintering for the analyses, we have decided to delete this piece of unused information from the manuscript (same argumentation as for the deletion of figure 3).

REVIEWERS' COMMENTS:

Reviewer #1 (Remarks to the Author):

Authors have addressed all my comments. I am satisfied with their responses.

Reviewer #3 (Remarks to the Author):

I don't have further comments. The authors have addressed the issues well.